# Exploring Strategies to Contact 3D Nano-Pillars

**DOI:** 10.3390/nano10040716

**Published:** 2020-04-10

**Authors:** Esteve Amat, Alberto del Moral, Marta Fernández-Regúlez, Laura Evangelio, Matteo Lorenzoni, Ahmed Gharbi, Guido Rademaker, Marie-Line Pourteau, Raluca Tiron, Joan Bausells, Francesc Perez-Murano

**Affiliations:** 1Institute of Microelectronics of Barcelona (IMB-CNM), CSIC, 08193 Barcelona, Spain; alberto.delmoral@imb-cnm.csic.es (A.d.M.); marta.fernandez@imb-cnm.csic.es (M.F.-R.); lauraevangelioaraujo@gmail.com (L.E.); matteo.lorenzoni@gmail.com (M.L.); joan.bausells@imb-cnm.csic.es (J.B.); francesc.perez@imb-cnm.csic.es (F.P.-M.); 2CEA-Leti, Département des Plateformes Technologiques, University Grenoble Alpes, 38000 Grenoble, France; ahmed.gharbi@cea.fr (A.G.); guido.rademaker@cea.fr (G.R.); marie-line.pourteau@cea.fr (M.-L.P.); raluca.tiron@cea.fr (R.T.)

**Keywords:** vertical pillars, AFM, HSQ, nano-indentation

## Abstract

This contribution explores different strategies to electrically contact vertical pillars with diameters less than 100 nm. Two process strategies have been defined, the first based on Atomic Force Microscope (AFM) indentation and the second based on planarization and reactive ion etching (RIE). We have demonstrated that both proposals provide suitable contacts. The results help to conclude that the most feasible strategy to be implementable is the one using planarization and reactive ion etching since it is more suitable for parallel and/or high-volume manufacturing processing.

## 1. Introduction

The electronics industry has extensively manufactured Metal Oxide Semiconductor Field-Effect-Transistor (MOSFET) devices by using planar topology for many decades. Scaling down the technology node beyond 32 nm has appeared problematic in terms of reliability, mainly because their performance is significantly degraded at these dimensions, e.g., by device variability and short channel effects. Both can be mitigated using 3D structures that present a better electrostatic control of the conduction over the device channel. In this scenario, FinFETs appeared as a suitable FET device, which is currently the basic building block for high volume manufacturing of integrated circuits [1]. Nevertheless, beyond 10 nm, an alternative FET architecture is the vertical nanowire, which can provide improved performance thanks to its full gate-all-around (GAA) structure and smaller footprint. Indeed, a FET based on a nanowire can be implemented in horizontal or vertical topologies [2]. While the former does not involve a relevant change in the manufacturing process flow, a vertical topology requires major changes in the manufacturing process [2]. In addition, vertical nanowires can also be used as building blocks for other devices, like single-electron transistors (SETs) or light emitters [3,4]. 

Although this type of device can involve many advantages in terms of area and performance (e.g., lower short channel effects), a more challenging manufacturing process is required, especially regarding procedures that create device contacts. Chemical mechanical polishing (CMP) [5] is the better-established method to planarize horizontal structures and then properly contact the devices. However, CMP is not fully adapted to 3D structures [6] and it becomes a drawback in some applications where the overall procedure must remain as simple as possible. To overcome this problem, we propose two different approaches. As a first option, we used local indentation by atomic force microscopy (AFM) that allows a nano-contact to be precisely placed on the top region of the pillar [7]. This method has been previously used to contact planar structures with high effectiveness [8,9]. Here, we explore the feasibility of extending this method to contact high aspect ratio vertical structures. The second option is based on the planarization strategy or etch-back process [10], using reactive ion etching (RIE). Thus, a relatively thick resist layer (several hundreds of nanometers) is deposited on top of the device; this layer is significantly thicker than the pillar, leaving the structure completely buried and only a topographic elevation can identify the position of the pillar. Later, we continuously thinned the layer by performing consecutive RIE cycles until the top face of the pillar is again exposed.

We have organized the article as follows: Section 2 explains in more detail both contacting methodologies and describes the samples, based on vertical nano-pillars, to be characterized. Section 3 shows the results of the electrical characterization of the contacted vertical nano-pillars. Section 4 presents the conclusions of the whole analysis.

## 2. Experimental Section

### 2.1. Device Concept and Fabrication Process

Figure 1a presents the concept of a single electron transistor based on a vertical nano-pillar. It consists of a stack of Si/SiO_2_/Si, with a unique silicon nanocrystal embedded into the SiO_2_ layer [3]. The silicon nanocrystal forms a quantum dot, in which energy levels are tuned with the gate metal electrode. The pillar height is ~56 nm, where the embedded thickness of the SiO_2_ layer is ~6 nm. Pillar diameters range from 30 to 100 nm. More details about the expected performance of this device structure can be found in [11].

Figure 1b shows a Scanning Electron Microscope (SEM) image of these pillars, manufactured by electron beam lithography (EBL). Alignment marks in close to the pillars are defined to easily locate the pillars position for the AFM nano-indentation, and the EBL process used to create the top metal electrodes. Figure 1c,d schematize the two approaches for fabricating a top metal contact to the nano-pillars that we have investigated. 

### 2.2. Planarization

Both contacting strategies are based on the use of an insulating bilayer to obtain a planar film of the desired thickness. As a polymeric resist, we have selected Shipley S1805 (Rohm and Haas, Paris, France), which is a positive resist for photolithography, and it can form thin films in a wide thickness range. Figure 2a shows the thickness of the S1805 layer as a function of solution concentration and spin velocity. A controlled thickness layer ranges from 600 nm with the original resist solution, down to 15 nm by high dilution (1:15) using MIBK (Merk, Darmstadt, Germany). Note that to mitigate the possible leakage current which can appear with the sole use of S1805, due to its poor electrical insulating properties, a bi-layer configuration is proposed. Therefore, before S1805 deposition, a ~15 nm layer of hydrogen silsesquioxane (HSQ) is deposited. HSQ is a well-established negative tone resist for EBL [12]. HSQ allows forming ultra-thin layers (tens of nanometers) with very low roughness and low dielectric constant (<3). Figure 2b shows that the HSQ layer thickness (Dow Corning XR-1541 2%) ranges from 35 nm to ~8 nm, as a function of concentration and spin speed. After deposition, an HSQ layer is converted into SiO_x_ by annealing at high temperatures (> 650 °C [12]) or by exposure to an electron beam [13]. In our case, the HSQ annealing consists of a 4-min bake at 80 °C. Afterward, to convert to SiO_x_ and obtaining an insulating layer, a long (1 h) annealing at 800 °C in a N_2_ environment is performed.

For creating a nano-contact via AFM indentation (Figure 3a), it is important to tune the stiffness of the resist. Therefore, the S1805 resist layer is hardened after deposition by performing two consecutive bakes. First, we baked the sample for 30 s at 90 °C and then we baked it again for 30 min at 130 °C. Planarization using RIE (Figure 4a) consists of etching-back the resist film by consecutive O_2_ plasma exposures, where the gas flow is kept constant at 50 sccm, and the power is shifted from 300 to 1000 W, depending on the required level of etching. 

It is also important to remark that to be able to contact a nano-pillar is crucial to obtain very smooth surfaces. For this reason, we monitor the surface roughness of both deposited layers (S1805 and HSQ) using AFM. Although both layers present a very low roughness, the HSQ layer shows the lowest, 0.25 nm rms, compared to 0.43 nm rms for S1805. 

### 2.3. AFM Nano-Indentation and Characterization

The AFM equipment employed was an ICON system with Nanoscope V electronics (BRUCKER, Camarillo, CA, USA). For AFM characterization, the AFM probes employed were Olympus AC160TS-R3 with an elastic constant of k~26 N/m and a Si tip. The surface is imaged in tapping mode, and the built-in capability of the system for performing nano-indentation is used. AFM current measurements were performed using SCM probes (k~0.2 N/m, Pt/Ir coated silicon tips) in contact mode (BRUKER, Camarillo, CA, USA). For detecting the current, a DC sample bias of –5 V was applied at a set point force of ~40 nN. A current amplifier of gain 1 V/nA connected to the tip was used. 

### 2.4. Definition of Metal Contacts

Once the pillar surface was revealed, electron beam lithography was used in both cases to define the contact area of the electrodes in the final step of contact fabrication. A 20 nm metal bi-layer of Cr-Au was deposited using e-beam evaporation and a lift-off process was performed to define the pads and lines for the electrical contacts. The manufactured contact dimensions were 50 × 50 µm^2^. 

## 3. Realization of Top Metal Contacts

In this section, we present the results obtained for contacting the top of the vertical nanowires using the two strategies.

### 3.1. Contacting by AFM Nano-Indentation

Although conductive AFM (c-AFM) is a suitable method to measure the electrical properties of individual structures [14], it is very time consuming and it cannot be used to characterize complete integrated circuits. Nevertheless, the methodology that we present here is interesting for contacting single pillars in complex integrated circuits, in combination with standard fabrication methods. The procedure is as follows: we used AFM nano-indentation to make an opening in the polymer on top of the pillar. By detecting the deflection of the AFM cantilever (Figure 3b), it was verified that the top of the pillar was reached. The lower part of the opening defines the contact area so that it has the same size as the diameter of the tip. The final metallic contact was made by a combination of electron beam lithography, metal deposition, and lift-off. Although the size of the contact area was usually the same, since it was determined by the tip diameter, the main cause of irreproducibility was the difficulty of aligning the opening with the pillar. However, in the absence of thermal drift during the AFM operation, this was successfully achieved approximately 80% of the time. Figure 3a depicts the indentation method by AFM. In detail, the HSQ layer was ~15 nm thick, and the S1805 layer was ~50 nm thick, so the pillar was covered by a ~10 nm resist. Then, by applying the required normal force, the desired tip indentation depth was achieved. Experimentally, we have found that to contact through 10 nm of baked S1805, the maximum force required was around 1000 nN. To ensure that the tip reached the top region of the pillar during the indentation, we acquired force-distance (F-d) curves, similar to what is shown in Figure 3b. It was possible to distinguish the first plastic deformation during early indentation and then, as the tip reached the hard substrate, the final part of the approach curve, which is linear (pure cantilever bending). The change in slope marked the moment when the elastic bending initiated, approximately defining the film thickness. Thanks to the designed alignment marks, the nano-pillars were properly located with AFM. Figure 3c,d presents AFM images as an example of a contact hole topography on a pillar. Each contact is individually defined. Figure 3e shows the obtained electrical characteristics of pillars that were 30 nm and 100 nm in diameter. Regarding the level of current obtained for each contacted pillar, we can conclude that a correct contact was achieved for each pillar, since the current was properly scaled as a function of the different pillar diameters. Additionally, to characterize the leakage current that flowed through the resist bi-stack, an electrical contact using the same procedure was used on an area without pillars. Thus, the observed negligible current level, in contrast to the indented curves, proved the validity of the proposal. Note that the signal was noisy for the non-pillar and 30 nm pillar scenarios, which can be related to the set-up used for the electrical characterization that was not optimized for current levels below 1 pA. Figure 3f presents the layout performed by EBL to contact all pillars used in this study.

### 3.2. Pillars Contacted by Etch-Back Process

We explored a second strategy to contact a vertical nano-pillar in a scenario more compatible with high-volume manufacturing. Similar to the nano-indentation approach, we first created a bi-layer stack to form an insulating film. In contrast to the AFM indentation process, the S1805 layer was significantly thicker (~600 nm), to completely bury the pillars for obtaining a planar film without any topography relief that revealed the pillar location. At this point, we applied consecutive O_2_ plasma exposures to uniformly etch-back the resist layer until the top part of the pillar was revealed again. Figure 4a depicts the schematic of the etch-back process using O_2_ plasma RIE. Figure 4b shows an SEM image of the top pillar region when it was exposed after consecutive O_2_ plasma cycles. This plasma sequence can be defined by different RIE parameters (e.g., power, flow and time). Nevertheless, our main aim was to obtain a final layer that was as flat as possible. For this, we investigated optimal O_2_ plasma power. While the use of high power during the plasma etching increased the etch rate, layer roughness increased as well, avoiding the proper realization of the final contacting step. We measured the etch rate per minute and the roughness of S1805 layers as a function of the applied power (150–1000 W) during the RIE plasma etching process. Figure 4c shows larger values for etch rate and roughness when the highest power was applied. In conclusion, consecutive O_2_ plasma cycles with low power allowed us to obtain smoother layers with a precise control of the final layer thickness, but required longer process times as the etch rate was decreased. Finally, once the top of the pillar was revealed, the contact pad was defined using EBL lithography with a later lift-off procedure. Figure 4d shows the current level that flowed through the larger and smaller contacted nano-pillars. Note that the current level was similar (in the range of few pA at 1 V bias) to the previously obtained contacts using AFM nano-indentation. It is worth noting that this procedure allows us to contact all the pillars at the same time, involving a relevant saving of time in comparison to the previous method, which is more focused on contacting a single nano-pillar. This fact enables this contacting methodology to be a more standardized concept for mass production.

To test the pillar surface quality resulting from the etch-back procedure, we produced conductive AFM (c-AFM) images of revealed Si nano-pillars, without the embedded SiO_2_ layer barrier and without the top metal contact. Si pillars without embedded oxide were more suitable to test the pillar contact quality because they can provide larger current levels and better c-AFM imaging. If proper current conduction through the silicon nano-pillar was observed during c-AFM, we verified the complete elimination of the resist without having compromised the quality of the top silicon. Figure 5a,b depicts a schematic of the measurement when the metallized tip contacts the top region of the pillar. Note that the estimated contact radius during c-AFM (and I-V curves) is always smaller than the pillar diameter. Figure 5c,d shows topography and corresponding current images of 100 nm diameter pillars in isolated configuration (c) and in an array configuration (d). The pillars are scanned at a 3-V bias. The current images show that current flows only in the areas corresponding to the revealed pillars, with values of hundreds of nA, approximately 10 times larger than what was measured on contacted pillars that included the embedded oxide layer. The fact that the contact area was smaller than the pillar diameter, along with the fact that we observed that the detected current scaled with pillar diameter, allowed us to conclude that the main resistance to current flow was provided by the pillar and not by the tip-pillar contact. 

Figure 6 shows I-V curves on several Si pillar of different diameters (30–100 nm) showing a Schottky diode behavior, as expected for a metal-semiconductor contact. For the larger pillar diameter, a higher current is obtained. Moreover, we have calculated the resistance of the pillars as a function of 1/d^2^, as shown in Figure 6b. The resistance is calculated from the derivate of the I/V curve in its linear part in its positive voltage. A correct scalability of the values is observed by proving the achievement of a correct contact of the nano-indented pillars. Note that the error bars show low variations for the different measurements done in this section. We have analyzed the possible influence of the contact resistance in our measurements, and we observed that it is not significant to take into account the Sharvin law [15]. 

## 4. Conclusions

Two strategies to contact ultra-small pillars (<100 nm) have been analyzed in terms of feasibility and possible manufacturing, i.e., AFM nano-indentation and etch-back planarization by O_2_ plasma RIE. In conclusion, we properly contacted pillars down to a 30-nm diameter using both methodologies. It is important to remark that we have been able, for the first time, to contact adequately vertical 30 nm diameter pillars using AFM nano-indentation. Similar currents through the pillars (pA at 1 V bias), and well-scaled as a function of pillar dimensions, are obtained for both contacting strategies, proving the viability of both methods. The indentation method is suitable when single/few pillars have to be contacted (faster processing). The etch-back methodology is more suitable for mass production fabrication as it enables to contact all the manufactured pillars at the same time.

## Figures and Tables

**Figure 1 nanomaterials-10-00716-f001:**
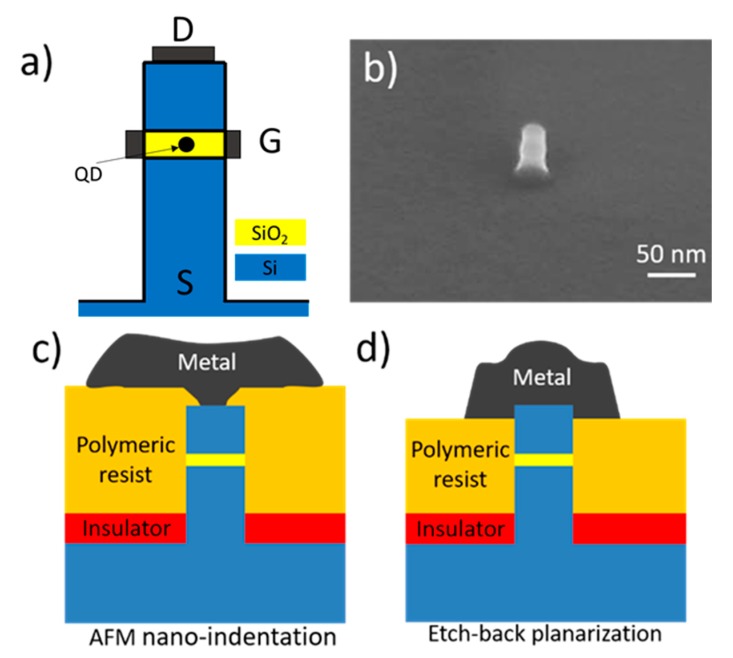
(**a**) Schematics of the concept based on vertical silicon nano-pillar: D: drain metal contact. G: gate metal contact. S: Source. QD: quantum dot embedded in a thin silicon layer. (**b**) SEM image of a vertical nano-pillar with the embedded silicon oxide layer (without metal contact). (**c**) Schematics showing a top metal contact performed by means of AFM nano-indentation. (**d**) Schematics of a top-metal contact performed by resist back-etching.

**Figure 2 nanomaterials-10-00716-f002:**
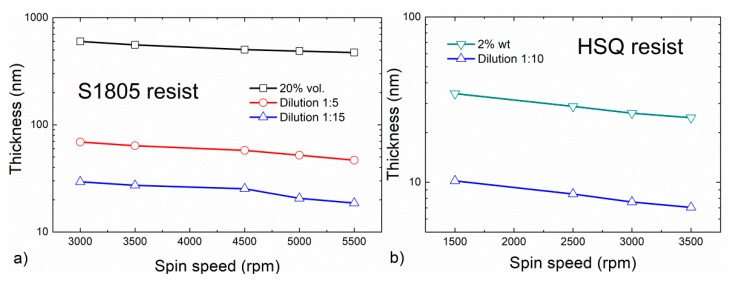
Layer thickness obtained for the different resist materials, Shipley S1805 (**a**) and HSQ (**b**), used along this work, for different dilutions.

**Figure 3 nanomaterials-10-00716-f003:**
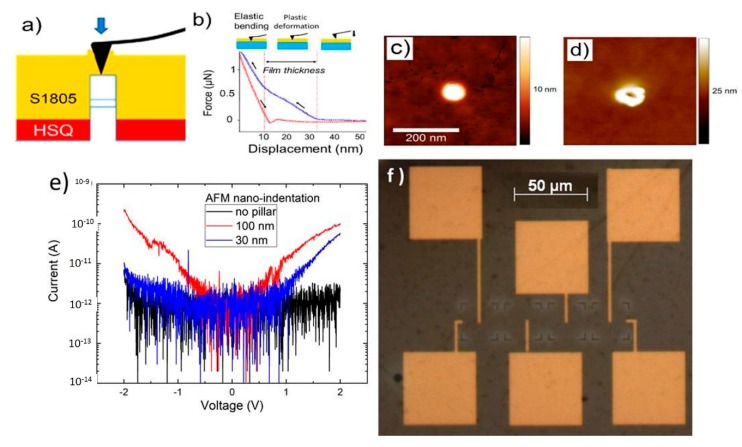
(**a**) Schematic of the AFM nano-indentation procedure. (**b**) Force distance curve acquired during indentation. (**c**) AFM image of a resist coated 30-nm diameter pillar before performing the nano-indentation. (**d**) AFM image of the same pillar after performing nano-indentation. (**e**) I-V curves after contacting two pillars, with diameters of 30 nm (blue) and 100 nm (red) by AFM indentation. In black, we show an I-V curve performed on a metal contact on top of an area without nano-pillar. (**f**) Optical image of contacted nano-pillars, pillars with different diameters are contacted, ranging from 100 nm (left) to 30 nm (right).

**Figure 4 nanomaterials-10-00716-f004:**
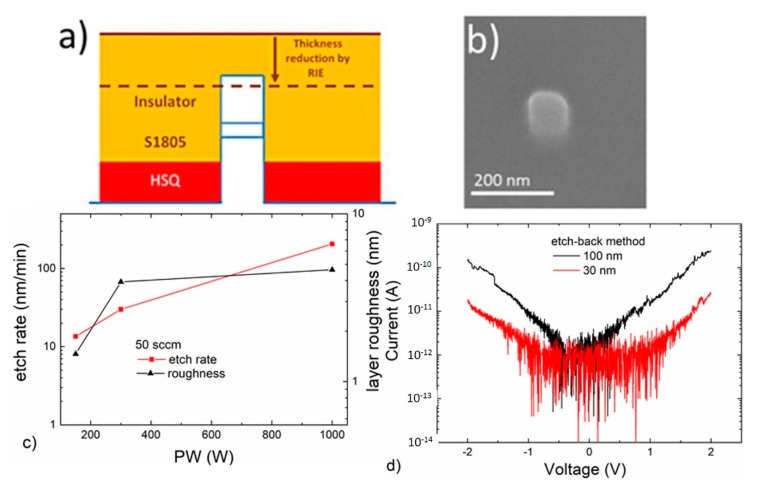
(**a**) Schematic for the etch-back (RIE planarization) method. (**b**) SEM image where the top region is shown above the S1805 resist layer. (**c**) etch-rate and root mean square (RMS) roughness of the S1805 resist layer after O_2_ plasma procedures. (**d**) I-V curves after contacting the pillars (30 and 100 nm) using the etch-back procedure.

**Figure 5 nanomaterials-10-00716-f005:**
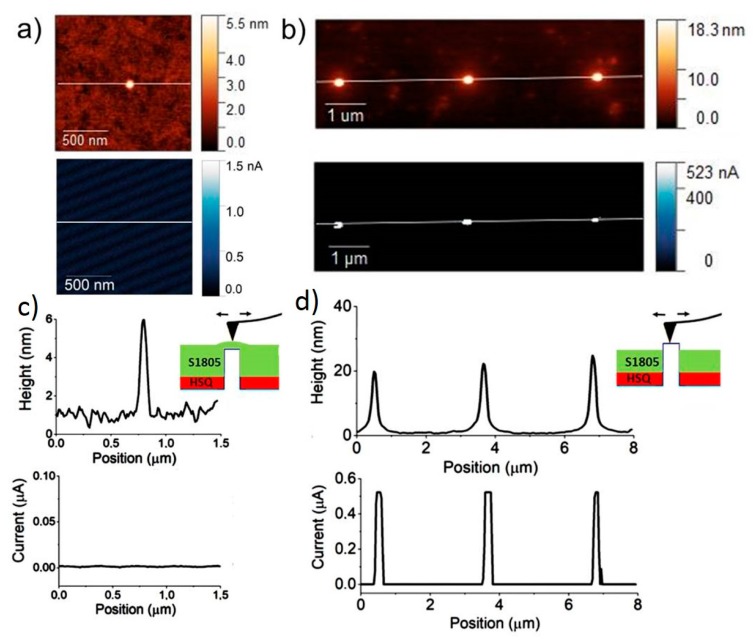
(**a**) Schematic of c-AFM measurement and (**b**) contact geometry on top of uncovered pillars; note that *D* is the pillar diameter, and *d* is the deformation due to contact in the z direction and a is the estimated contact radius. In panels (**c**) and (**d**) c-AFM images (topography and current) of 100 nm diameter pillars acquired at 3 V bias are shown. (**c**) Images and corresponding height profiles of a single pillar before etch-back, since the pillar is buried into the resist, no current is detected (**d**) images and corresponding profiles of three revealed pillars after etch-back, current is measured on top of each pillar.

**Figure 6 nanomaterials-10-00716-f006:**
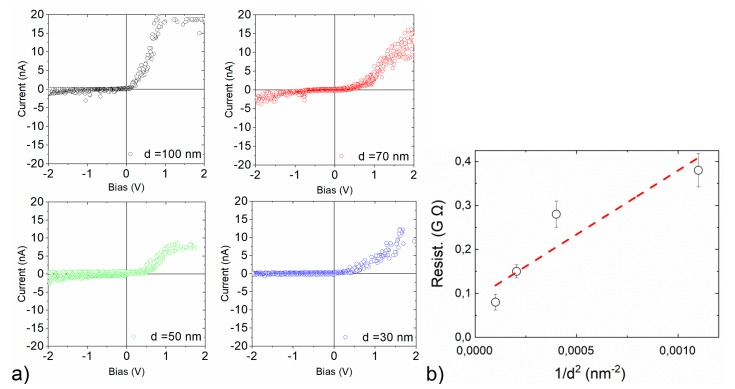
(**a**) I-V characterization of Si nano-pillars with different diameters (30–100 nm), ±2 V bias sweep. Force was kept constant at 50 nN during the bias sweep. I-V shape recalls the behavior of a Schottky diode, as expected for a metal-semiconductor contact. (**b**) Resistance of the nano-pillars obtained from the measured current. The resistance is calculated from the derivate of the I/V curve in the linear part of the curve for positive voltage. We have introduced error bars to show the reproducibility of all measurements.

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
