# Peer review of "Exploring Strategies to Contact 3D Nano-Pillars"

_nanomaterials, 2020, doi:10.3390/nano10040716_

Round 1

Reviewer 1 Report

In this article, the authors present two strategies to realize electrical contact for nano-pillars with 30-100 nm diameters in thin film semiconductor devices. In section 3.1, electrical contact to the top of the pillar embedded in the insulator layer was achieved by AFM nano-indentation. In section 3.2, the exposure of the pillar head by etch-back process was performed using O2 plasma RIE. The nano-pillar-based devices prepared in the present study exhibited appropriate I-V characteristics. Both approaches are interesting for the fabrication of nano-electronic devices with vertical topologies. I think that this paper can be accepted for publication in Nanomaterials as it is. It is recommended to add a label to the vertical axis of Figure 3e.

Author Response

First of all, we would like to thanks the reviewer for its valuable comments, what has helped us to enhance our contribution to the Nanomaterials Special Issue.

Reviewer 1

In this article, the authors present two strategies to realize electrical contact for nano-pillars with 30-100 nm diameters in thin film semiconductor devices. In section 3.1, electrical contact to the top of the pillar embedded in the insulator layer was achieved by AFM nano-indentation. In section 3.2, the exposure of the pillar head by etch-back process was performed using O2 plasma RIE. The nano-pillar-based devices prepared in the present study exhibited appropriate I-V characteristics. Both approaches are interesting for the fabrication of nano-electronic devices with vertical topologies. I think that this paper can be accepted for publication in Nanomaterials as it is. It is recommended to add a label to the vertical axis of Fig. 3.e.

Thanks to the reviewer’s comment, then, we have improved our contribution by modifying Figure 3.e vertical axis.

Reviewer 2 Report

The authors report two electrical contacting methods for nano-pillars. The ms can be accepted for publication in nanomaterials, provided the authors address the following shortcomings. 

Authors must comment on noise in fig 3b and 4d. What is the origin of the noise? I wonder if the measurment shown in fig 3b is reproducible for another pillars. Authors must provide more measurements to clarify this, otherwise the data cannot be considered credible. Also, I wonder if the indentation method will produce the same contact area every time, probably not. Authors must clarify this issue as they claim novelty for this contacting method in the conclusion (L182). Same applies to fig 4d. More than one measurement is required to prove reproducibility. Authors must add error bars to the data in fig 6b. also, I do not understand why there are two fits shown.

Author Response

First of all, we would like to thanks the reviewer for its valuable comments, what has helped us to enhance our contribution to the Nanomaterials Special Issue.

Reviewer 2

The authors report two electrical contacting methods for nano-pillars. The ms can be accepted for publication in nanomaterials, provided the authors address the following shortcomings. 

Authors must comment on noise in Fig. 3.b and 4.d. What is the origin of the noise?

The observed noise can be related to the set-up used for the electrical characterization of the devices, which is not optimized for current levels below 1 pA. We have introduced into the text a short explanation to clarify this effect. See comment R2.1.

I wonder if the measurement shown in Fig. 3.b is reproducible for other pillars. Authors must provide more measurements to clarify this, otherwise the data cannot be considered credible. Also, I wonder if the indentation method will produce the same contact area every time, probably not. Authors must clarify this issue as they claim novelty for this contacting method in the conclusion (L182).

We have clarified the reproducibility in the revised manuscript, see comment R2.2. The size of the contact is determined by the tip diameter. The main source of non-reproducibility comes from the difficulty in aligning the nano-indentation with the pillar, but by minimizing the thermal drift during the AFM operation, successful contacts are obtained in approximately an 80% of the occasions.

Same applies to Fig. 4.d. More than one measurement is required to prove reproducibility. Authors must add error bars to the data in Fig. 6.b. also, I do not understand why there are two fits shown.

After the reviewer suggestion we have included in Fig. 6b error bars for each point; then, we have removed the second red line to avoid possible misunderstandings on the figure. See comment R2.3.

Reviewer 3 Report

The manuscript by Francesc Perez et al. discusses the methods of preparation of nano-pillared surfaces for further fabrication of FET and SET devices. Although it does not present interest from the view point of fundamental science, it is a fairly good account of approaches that may be of substantial merit for the methodology and technology development. The experimental details are well discussed and clearly presented. The work is original and and be of interest to the readers of Nanomaterials.  

I recommend the authors to improve the presentation. First, more care is to be given to abbreviations used in the text, in particular in the Intro. Second, hte figures are to be presented in better quality: resolution (figure 5), uniform font size (figure 3, 5), axis title (Figure 3e), more careful captioning for figure 6 b (what the two red lines mean?), etc. Third, the text is to be clarified or explained: "Note that the estimated contact radius during c-AFM (and I-V curves) is always smaller than the pillar diameter. Therefore, the current level should only be limited by the pillar diameter." - Lines 164-166. In the current form it is self-contradicting. Also: line 168 - correction is needed.

In summary, I recommend the manuscript for publication upon minor corrections of "cosmetic" nature related to presentation.

Author Response

First of all, we would like to thanks the reviewer for its valuable comments, what has helped us to enhance our contribution to the Nanomaterials Special Issue.

Reviewer 3

The manuscript by Francesc Perez et al. discusses the methods of preparation of nano-pillared surfaces for further fabrication of FET and SET devices. Although it does not present interest from the view point of fundamental science, it is a fairly good account of approaches that may be of substantial merit for the methodology and technology development. The experimental details are well discussed and clearly presented. The work is original and be of interest to the readers of Nanomaterials.  

I recommend the authors to improve the presentation. First, more care is to be given to abbreviations used in the text, in particular in the Intro.

Thanks to the reviewer comment we have improved our contribution and we have taken care about the use of the abbreviation along the new contribution text.

Second, the figures are to be presented in better quality: resolution (Fig. 5), uniform font size (Figs. 3, 5), axis title (Fig. 3.e), more careful captioning for Fig. 6.b (what the two red lines mean?), etc.

Regarding the reviewer’s comment we have improved the quality of the figures. We have introduced the title of the vertical axis of Fig. 3e. We have enhanced the quality of Fig. 5. To avoid misunderstanding we have modified Fig. 6.b by including error bars for each point and its figure caption. See comment R3.2.

Third, the text is to be clarified or explained: "Note that the estimated contact radius during c-AFM (and I-V curves) is always smaller than the pillar diameter. Therefore, the current level should only be limited by the pillar diameter." - Lines 164-166. In the current form it is self-contradicting. Also: line 168 - correction is needed.

We have clarified the text as suggested by the reviewer (Comment R3.3):  the fact that the contact area is smaller than the pillar diameter, along with the fact that we observe that the detected current scales with the pillar diameter allows us to conclude that the main resistance to current flow is provided by the pillar and not by the tip-pillar contact.

Round 2

Reviewer 2 Report

The authors have complied with most of the issues raised in the three review reports. However, after the last revision a new peculiarity appeared. In Figure 6 they now show the I-V characteristics for nano-pillars. The characteristics exhibit non-linear diode-like behavior. This is fine, but then the authors plot the resistance vs 1/d^2 in fig 6b. Since the characteristics by nature are non-linear, the authors must indicate which part of of the I-V curve is used to obtain the resistance. Hence, strictly speaking they have not provided convincing experimental evidence that the measured characteristic is not influenced by the contacting method, and that the contact area is reproducible. Also, it can be argued that the "linear" scaling in fig 6b is not very linear, as it could be fitted to many other functions with comparable sum of residuals. Authors must explain this much better or adjust their claims. 

Author Response

First of all, we would like to thanks all the reviewers for their comments and suggestions always looking for enhance our contribution. Then, next we would like to answer their comments and questions:

Reviewer 2

The authors have complied with most of the issues raised in the three review reports. However, after the last revision a new peculiarity appeared. In Figure 6 they now show the I-V characteristics for nano-pillars. The characteristics exhibit non-linear diode-like behavior. This is fine, but then the authors plot the resistance vs 1/d^2 in fig 6b. Since the characteristics by nature are non-linear, the authors must indicate which part of of the I-V curve is used to obtain the resistance. Hence, strictly speaking they have not provided convincing experimental evidence that the measured characteristic is not influenced by the contacting method, and that the contact area is reproducible. Also, it can be argued that the "linear" scaling in fig 6b is not very linear, as it could be fitted to many other functions with comparable sum of residuals. Authors must explain this much better or adjust their claims. 

In order to clarify the reviewer’s doubt we would like to note that the resistance is calculated from the derivate of the I/V curve in its linear part in its positive voltage. In this sense, we have included a new explanation in the main text and in the Fig. 6b caption, as well. See comment R2.1.